∂ | **Open Peer Review** | Antimicrobial Chemotherapy | Research Article

# Metabolomics insights into the polyketide-lactones produced by *Diaporthe caliensis* sp. nov., an endophyte of the medicinal plant *Otoba gracilipes*

Esteban Charria-Girón,[1,2,3] Yasmina Marin-Felix,[1,2] Ulrike Beutling,[4] Raimo Franke,[4] Mark Brönstrup,[4] Aida M. Vasco-Palacios,[5,6] Nelson H. Caicedo,[3,7] Frank Surup[1,2]

**ABSTRACT** Plants of the genus *Otoba* have been the basis for the treatment of tropical diseases in indigenous communities of countries like Colombia. Despite the lack of knowledge about their bioactive principles, endophytic fungi derived from medicinal plants are a prolific source of innovative chemistry. We systematically investigated the secondary metabolite production of a previously undescribed species of *Diaporthe*, herein introduced as *Diaporthe caliensis* sp. nov., using different metabolomics approaches together with classical chemical screening. To get an outline of the chemical space produced by this fungus, an exploratory molecular networking (MN) analysis was undertaken. A major molecular family was found to contain the known 10-membered lactone phomol (**1**), together with other putative congeners as compound **3**. After isolation by preparative high-performance liquid chromatography, we confirmed phomol (**1**) as the main reason for the antimicrobial activity of the crude extract. The unknown absolute configuration of **1** was determined by the synthesis of α-methoxy-α-trifluoromethylphenylacetyl (MTPA)-esters and chemical degradation experiments. Moreover, caliensolides A (**2**) and B (**3**) were isolated, and their structures were elucidated as novel butenolides structurally unrelated to **1**. Overall, the initial MN analysis incorrectly clustered compounds **1** and **3** within a single molecular family, despite evident differences in chemical structures and biosynthetic origin. Contrariwise, the unsupervised substructure discovery algorithm MS2LDA provided a deeper understanding of the fragmentation patterns and correctly clustered the polyketide-lactones produced by *D. caliensis* sp. nov. Our findings encourage the exploration of Colombian fungal diversity, which as demonstrated here could result in the discovery of new natural products.

**IMPORTANCE** The integration of metabolomics-based approaches into the discovery pipeline has enabled improved mining and prioritization of prolific secondary metabolite producers such as endophytic fungi. However, relying on automated untargeted analysis tools might lead to misestimation of the chemical complexity harbored in these organisms. Our study emphasizes the importance of isolation and structure elucidation of the respective metabolites in addition to deep metabolome analysis for the correct interpretation of untargeted metabolomics approaches such as molecular networking. Additionally, it encourages the further exploration of endophytic fungi from traditional medicinal plants for the discovery of natural products.

**KEYWORDS** antibiotics, metabolomics, natural products, secondary metabolites, structure elucidation, NMR, dereplication

Address correspondence to Yasmina Marin-Felix, yasmina.marinfelix@helmholtz-hzi.de, or Frank Surup, frank.surup@helmholtz-hzi.de.

The authors declare no conflict of interest.

See the funding table on p. 14.

Colombian indigenous communities have traditionally employed medicinal plants to treat infectious tropical diseases (1–4). The genus *Otoba* (family Myristicaceae) comprises 12 accepted species native throughout the humid northern Neotropics (5). The species within this genus are well known in Latin American traditional medicine for their use against chagas, leishmaniasis, malaria, and fungal and mite infections among other applications (1–3, 6–8). *Otoba gracilipes* is one of the native medicinal plants from the Valle del Cauca River geographic area, recently considered endangered (status: near threatened; 9), and from which no information about its bioactive principles has been ever reported (10–12). Regrettably, medicinal plants and derived ethnomedicines have not been systematically studied to understand the molecular mechanisms and benefits of their bioactive components (2).

In a similar manner, fungal endophytes isolated from medicinal plants have also shown their ability as producers of bioactive compounds and accordingly their potential application in agriculture, pharmaceutics, and biocontrol, among other applications (13–16). In a preceding study, endophytic fungi were isolated from the medicinal plant *Otoba gracilipes* in Colombia and studied for their antimicrobial potential (12). Among them, an isolate of *Diaporthe*, which is here proven to represent a new species based on molecular data, was found to exhibit prominent antimicrobial effects and was prioritized for a metabolomics MS/MS-based investigation of its secondary metabolome in combination with a classical chemical screening. Herein, we present the description of the new species *Diaporthe caliense* sp. nov., a systematic capture of its secondary metabolome, as well as an account of the biological properties of the isolated metabolites.

## RESULTS

### Molecular phylogeny and taxonomy

The lengths of the fragments of the phylogenetic inference for the five loci used in the combined data set were 572 bp (ITS), 449 bp (*cal*), 373 bp (*his3*), 452 bp (*tef1*), and 862 bp (*tub2*), comprising in total 99 taxa. The length of the final alignment was 2,708 bp. Figure 1 shows the consensus maximum likelihood (ML) tree, including bootstrap support (bs) and Bayesian posterior probability (pp) values at the nodes. Our strain was located in an independent branch distant from other species of *Diaporthe*, demonstrating that this represented a new species, which is introduced herein as *D. caliensis* sp. nov. Unfortunately, the new species lacked sporulation in all media tested in the present study. Therefore, the introduction of it is based on molecular data only. The alignment of each locus is available in the supplemental material (Table S5).

### *Diaporthe caliensis* Y. Marín, sp. nov. MycoBank MB845610

### *Etymology*

The name refers to Cali, the city where this fungus was collected.

### *Type material*

Colombia, Cali, from the leaves of *Otoba gracilipes*, November 2019, Nelson H. Caicedo (holotype HUA230971, ex-type CM-UDEA-H27, culture isotype CBS 149729 = STMA 22040).

### *Not sporulated*

*Diaporthe caliensis* differs from its closest phylogenetic neighbor, *Diaporthe phaseolorum* by unique fixed alleles in four loci based on alignments of the separate loci included in the supplementary material: ITS positions 159 (G); *his3* positions 135 (T), 136 (T), 137 (T), 138 (T), 139 (C), 140 (A), 141 (C), 142 (A), 143 (Y), 144 (C), 145 (C), 146 (A), 147 (C), 148 (C), 149 (C), 150 (A), 151 (A), 152 (A), 153 (T), 154 (C), 155 (A), 156 (A), 157 (T), 158 (C), 159 (A), 160 (A), 161 (C), 162 (T), 163 (T), 164 (C), 165 (A), 166 (C), 167 (C), 168 (C), 169 (T), 170 (C), 171 (G), 172 (T), 173 (T), 174 (T), 175 (A), 176 (C), 177 (C), 178 (C), 179 (T), 180 (G), 181 (C),

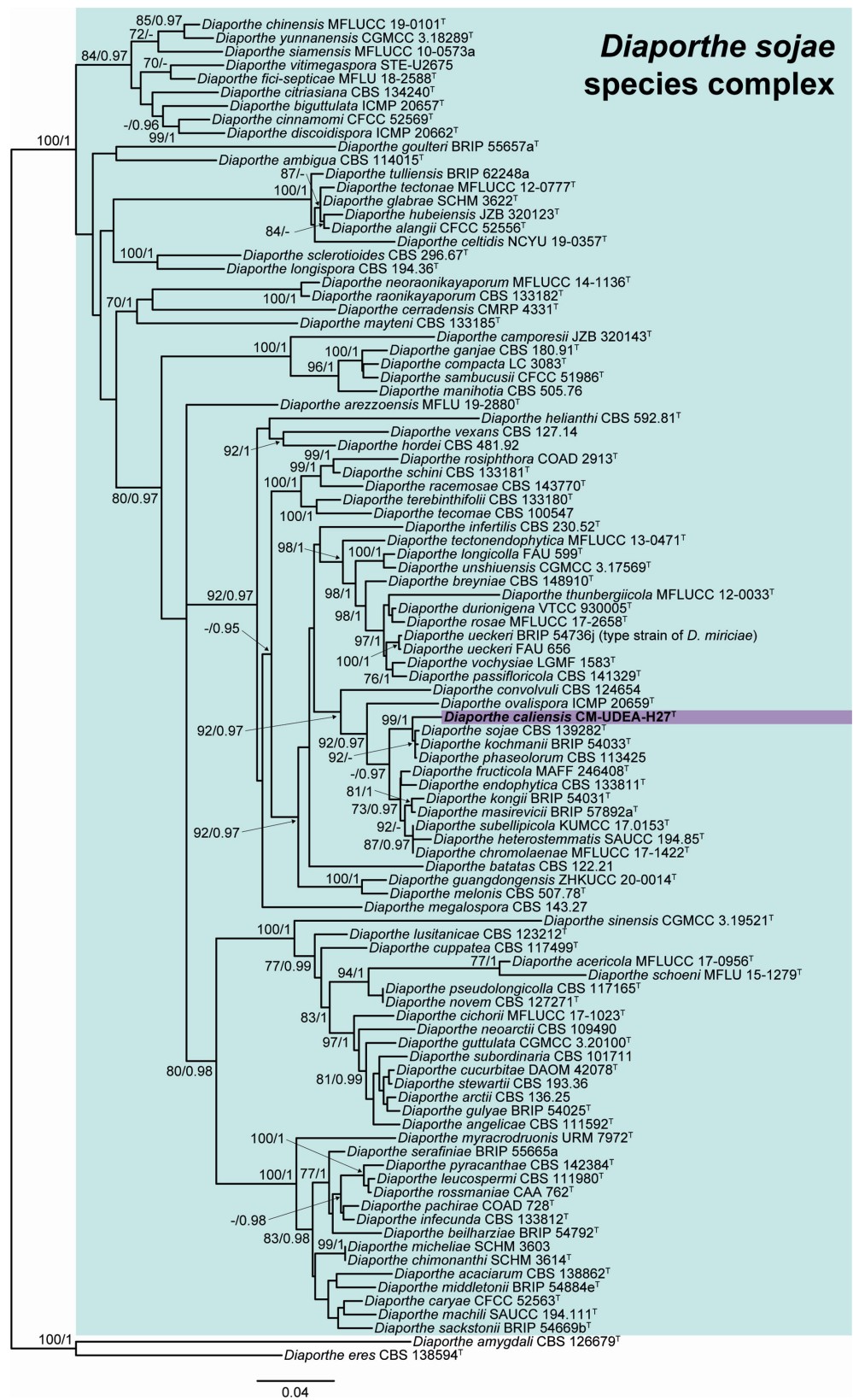

**FIG 1** RAxML phylogram obtained from the combined ITS, *cal*, *his3*, *tef1*, and *tub2* sequences of our strain and related *Diaporthe* spp. *Diaporthe amygdali* CBS 126679[T] and *Diaporthe eres* CBS 138594[T] were used as an outgroup. Bootstrap support values ≥ 70/Bayesian posterior probability scores ≥ 0.95 are indicated along branches. Branch lengths are proportional to distance. The new taxon is indicated in bold. Type material of the different species is indicated with a superscript T.

182 (T), 183 (G), 184 (A), 185 (C), 186 (C), 187 (G), 188 (T), 189 (C), 190 (G), 191 (C), 192 (C), 193 (T), 194 (C), 195 (T), 196 (T), 201 (C), 202 (C), 203 (T), 204 (C), 205 (C), 206 (C); *tef1* positions 189 (T), 302 (T), 308 (G), 337 (A), 338 (A), 339 (C), 340 (A), 341 (G), 342 (C), 343 (A), 344 (T), 345 (C), 346 (A), 347 (C), 348 (C), 349 (T), 350 (T), 351 (C), 352 (A), 353 (T), 354 (T), 355 (C), 356 (C), 357 (C), 358 (A), 361 (T), 362 (C), 363 (T), 364 (G), 365 (T), 366 (T), 367 (G), 368 (C), 369 (T), 370 (C), 371 (G), 372 (G), 373 (G), 374 (G), 375 (G), 376 (A), 377 (G), 378 (C), 379 (T), 380 (T), 381 (T), 382 (T), 383 (C), 384 (G), 385 (A), 386 (C), 387 (T), 388 (C), 389 (G), 390 (C), 391 (T), 392 (T); *tub2* positions 360 (C), 392 (C), 405 (indel), 443 (G), 564 (T), 567 (T), 787 (T), 820 (C), 862 (T).

### Culture characteristics

Colonies on potato dextrose agar (PDA) covering the surface of the Petri dish in 2 weeks were white to yellow-white (158B–C), cottony, raised, and margins were filamentous to fimbriate; reverse yellow-white (158A–B) with a ring grayed-green (197A) and center grayed orange (165B–C). Colonies on malt extract agar (MEA) covering the surface of the Petri dish in 2 weeks were white to grayed white (156A–B), cottony, raised, and margins were filamentous to fimbriate; reverse yellow white (158A–C). Colonies on oatmeal agar (OA) covering the surface of the Petri dish in 2 weeks were white to yellow-white (158A–B), velvety to cottony, flat to raised in some zones, and margins were filamentous to fimbriate; reverse white to yellow-white (158B–C) (Fig. S1).

### Notes

This species is introduced based only on molecular data since sporulation could not be induced in any media used. *Diaporthe caliensis* is located in the *Diaporthe sojae* species complex, which also includes *D. sojae* and *Diaporthe ueckeri*, both isolated in Colombia as our new species (17, 18). This species complex is the largest in *Diaporthe* and includes *D. phaseolorum* and *D. sojae*, which are important pathogens of beans and soybean, respectively (17). The latter species are the most related to *D. caliensis*, together with the moderate pathogenic species of sunflower *Diaporthe kochmanii* and the endophytic *Diaporthe ovalispora* (19, 20).

## Metabolomics exploratory analysis

To inspect the secondary metabolome of *D. caliensis* sp. nov., an untargeted metabolomics analysis was performed. The crude extract obtained after the solid fermentation of this fungus was analyzed by ultrahigh-performance liquid chromatography coupled with diode array detection and high-resolution electrospray ionization tandem mass spectrometry (UHPLC-DAD-HRESI-MS/MS). LC-MS/MS raw data were first pre-processed and the obtained feature table was dereplicated based on accurate *m/z*, MS/MS spectra, retention time, and UV/Vis spectra using the available libraries of metabolites reported under *Diaporthe* or *Phomopsis* (outdated name for the anamorphic state of *Diaporthe*) in the NP Atlas website (*Diaporthe* spp. = 112, *Phomopsis* spp. = 340, total = 452) and the Compound Crawler and MetFrag tools (21, 22). Subsequently, of the 2,078 features detected at the MS level and 491 features at the MS/MS level, only 11 features fit the criteria for annotation confidence based on mass accuracy (mass deviation < 2 mDa), isotope pattern (mSigma < 20), and MS/MS spectra (MS/MS score > 900) matching (Table S2). The number of annotated features accounted only for 0.5% and 2.2% of the detected features at the MS and MS/MS levels, respectively, highlighting the underrepresentation of metabolites produced by this fungus when compared to the reported natural products in the public databases.

Since the majority of features remained unannotated, we decided to use a feature-based molecular network (FBMN) approach using the Global Natural Products Social Molecular Networking (GNPS) infrastructure to gain a broader overview of the observed chemical space of *D. caliensis* sp. nov. (23, 24). The obtained FBMN was further analyzed with the MolNetEnhancer program to propagate chemical class annotations to the full subnetwork. The 491 features detected at the MS/MS level were organized into 44

molecular families (MFs) with at least two clustered nodes and 180 features represented as singletons (Fig. 2A).

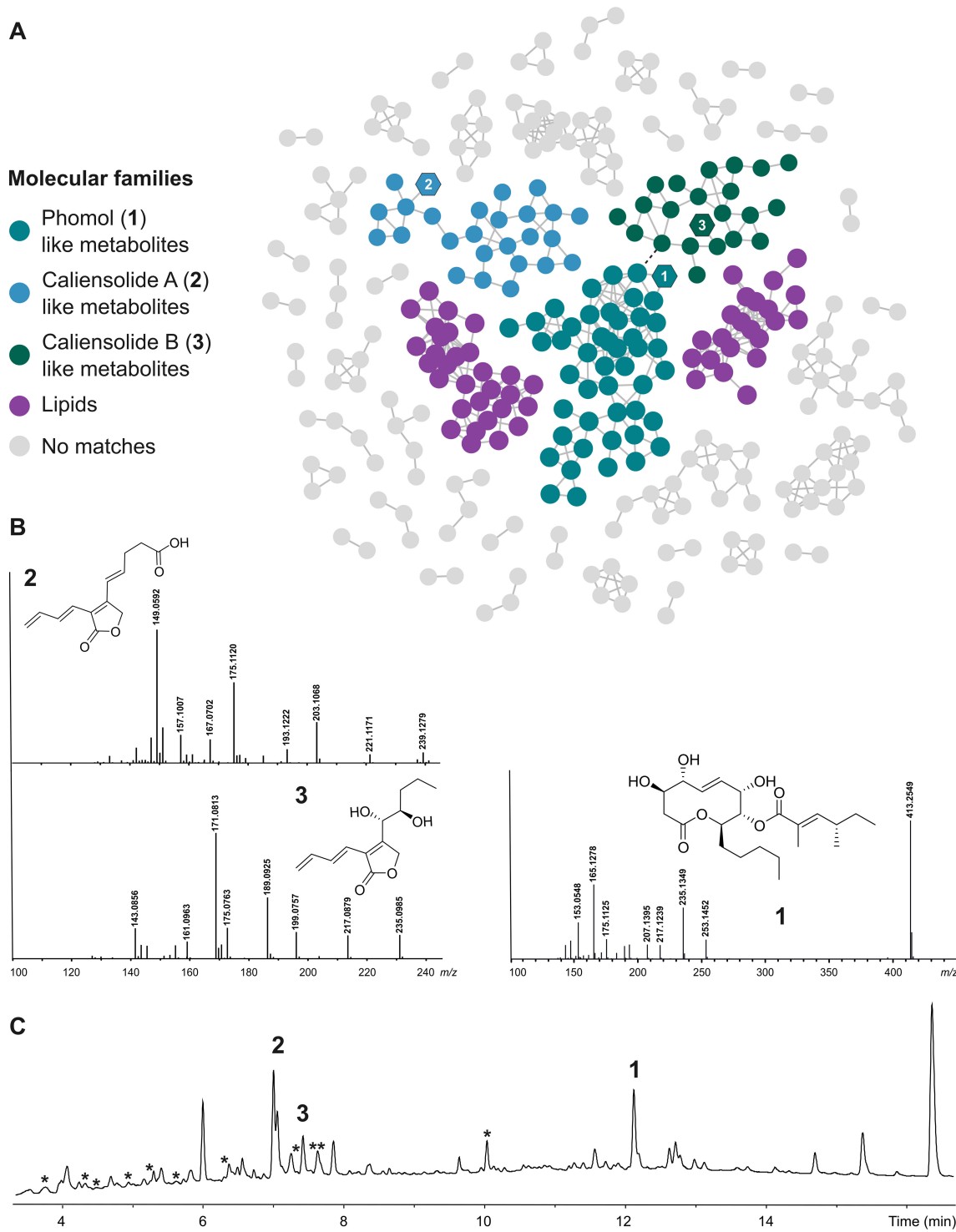

FIG 2 (A) Feature-based molecular network obtained from the crude extract of the oatmeal culture of *D. caliensis* sp. nov. with the molecular families annotated by MolNetEnhancer and isolated metabolites represented by hexagon-shaped nodes (cosine score = 0.5). Molecular families comprising the isolated metabolites were manually annotated in the MN. Compounds **1** and **3** were enclosed within the same MF in two different subclusters. (B) Reference MS/MS spectra of **1–3** isolated from the culture of *D. caliensis* sp. nov. (C) HPLC-UV/Vis chromatogram (210 nm) of the crude extract with isolated metabolites (**1–3**) and dereplicated metabolites indicated by bold numbers and asterisks, respectively, over the corresponding peaks.

Moreover, we identified that the biggest MF contained one of the annotated metabolites, phomol, a 10-membered polyketide-lactone originally found to be produced by an endophytic strain of *Diaporthe* (producer was referred to the outdated name *Phomopsis*) (25). This MF clustered 69 nodes with a precursor mass ranging from 235.0970 to 537.3429 Da. However, this MF appeared to be divided into two subclusters connected by two nodes presenting a cosine score of 0.8208, where the subcluster containing phomol was represented by molecules of relatively higher molecular weight (Fig. 2A).

## Isolation and structure elucidation disclose chemical diversity

Since the major MF detected during our exploratory FBMN analysis pointed toward the production of secondary metabolites related to the polyketide-lactone phomol, we embarked on the targeted isolation of the major metabolites found in the crude extract. Preparative HPLC afforded compounds **1–3** (Fig. 3); their structures were elucidated by HRESI-MS and 1D- and 2D-NMR spectroscopic data (Fig. S2 through S19).

Compound **1** was obtained as a colorless oil and its molecular formula was established as $C_{22}H_{36}O_7$ (five degrees of unsaturation) according to the molecular ion peak cluster at $m/z$ 435.2532 [M + Na] $^+$ in the HRESI-MS spectrum. Comparison of the $^1H$ and $^{13}C$ NMR data of **1** measured in DMSO-$d_6$ with those of Basnet et al. (26) confirmed its identity as phomol (25). We addressed the unknown absolute configuration of the aglycon by Mosher's method. The introduction of three units of the chiral auxiliary MTPA on the macrolide impedes simple use of the $\Delta\delta^{SR}$ chemical shift analysis developed for monofunctional compounds. Thus, the crossed effects between auxiliaries were taken into account, based on the evaluation of the anisotropic effects caused individually by every auxiliary linked to hydroxyl groups at C–3, C–4, and C–7, predicting the resulting combination of the individual effects. In particular, the observed positive $\Delta\delta^{SR}$ chemical shifts of 2–$H_a$, 2–$H_b$, 9–H, and 10–$H_2$, indicated a 3R,7S configuration. Thus, we assigned the absolute configuration of the aglycon as 3R,4R,7S,8S,9R. The absolute configuration of C–4′ in the side chain of **1** was determined by a degradation experiment. The $\Delta^{C2',C3'}$ double bond was cleaved via ozonolysis, followed by an oxidative workup (Fig. 4). The presence of (S)-2-methyl butyric acid was confirmed with authentic standards by chiral GC-MS analysis.

Compound **2** was obtained as a yellow oil. The molecular ion cluster at $m/z$ 239.1275 [M + H] $^+$ in the HRESI-MS spectrum indicated that the molecular formula is $C_{13}H_{18}O_4$ (five degrees of unsaturation). $^1H$ and HSQC NMR data of **2** indicated the presence of three olefinic methines, two oxymethines, one exomethylene, one oxymethylene, two further methylenes, and one terminal methyl group. The $^{13}C$ NMR spectrum revealed two additional sp$^2$ hybridized carbons without bound protons and one carboxylic acid. COSY NMR data connected these 1,3-butadienyl and 1,2-dihydroxy-pentyl fragments. Finally, HMBC correlations from 5–H to C–6, C–7, C–13; from 8–H to C–6, C–7, C–12 and from 13–$H_2$ to C–5, C–6, C–7 established the scaffold of **2** (Fig. S18). The large coupling constant $J_{H8,H9}$ = 15.6 Hz indicated an E configuration of the $\Delta^{8,9}$ double bond. Although a J-res analysis for the relative configuration yielded inconclusive results, the absolute configuration of **2** could be assigned by advanced Mosher's method. The observed

**FIG 3** Chemical structures of compounds **1–3**.

**FIG 4** Degradation strategy for phomol (**1**) to obtain 2-methyl butyric acid.

pattern of $\Delta\delta^{SR}$ chemical shifts with negative values for 5–H, 8–H, 9–H, 10–H, and 13–$H_2$ on the one, and positive values for 2–$H_2$, 3–$H_2$, and 4–H is diagnostic for a *4R,5S* configuration (Fig. S19) (27). We propose to name **2** as caliensolide A; the systematic name of **2** is 3-((*E*)-buta-1,3-dien-1-yl)–4-((1*S*,2*R*)–1,2-dihydroxypentyl)furan-2(5*H*)-one.

Compound **3** was obtained as a white powder and its molecular formula was established as calculated for $C_{13}H_{14}O_4$ (seven degrees of unsaturation) according to the molecular ion peak cluster at *m/z* 235.0970 [M + H] $^+$ in the HRESI-MS spectrum. The NMR data of **3** were highly similar to those of **2**, with the key difference being the exchange of the two oxymethines with a double bond and the terminal methyl with a carboxylic acid. The large coupling constant $J_{H4,H5} = 16.0$ Hz indicated an *E* configuration of the $\Delta^{4,5}$ double bond. Consequently, **3** was assigned as (*E*)−5-(4-((*E*)-buta-1,3-dien-1-yl)−5-oxo-2,5-dihydrofuran-3-yl)pent-4-enoic acid and named caliensolide B.

## Unsupervised analysis illustrates the complexity of polyketide-lactones produced by *D. caliensis*

Coming back to the results provided by the exploratory FBMN analysis, we identified that the isolated metabolites, phomol (**1**) and the caliensolides A and B (**2** and **3**), were distributed incorrectly into two different MFs. Compounds **1** and **3** were clustered within the same MF, which as previously stated appeared to be divided into two different subclusters (Fig. 2). This issue revealed the limitation of the FBMN approach and MolNetEnhancer to correctly cluster the observed chemical diversity and propagate the observed chemical class annotations. Consequently, we tried to assess the robustness of the FBMN approach by modifying the threshold for the cosine score, which determines whether an edge must be kept between two nodes. When the threshold was decreased, similar network topologies were obtained, yet without an MF containing both compounds **2** and **3**. In the opposite case when the cosine score was increased over 0.6828, the FBMN topology was altered, and as a result, compound **3** was then represented as a singleton (Fig. S20).

At this point, we used the unsupervised substructure discovery approach MS2LDA to annotate shared chemical motifs within the different fragmented MS features in our data set and evaluate if this approach might provide a clearer explanation of the observed phenomenon. MS2LDA is a topic modeling algorithm, which extracts biochemically relevant molecular substructures ("Mass2Motifs") from spectra as sets of co-occurring molecular fragments and/or neutral losses (28). The analysis allows us to group molecules based on the presence of molecular substructures, regardless of classical spectral similarity. Accordingly, we identified a subset of the FBMN consisting of nine different nodes related to the Mass2Motif 620, among which the caliensolides A and B (**2** and **3**) were also present (Fig. 5). The presence of this Mass2Motif confirmed the chemical similarity among the isolated caliensolide derivatives, although the MN did not result in a similar output. Also, the fact that compounds **1** and **3** were clustered within the same MF is explained by the findings obtained by the MS2LDA approach,

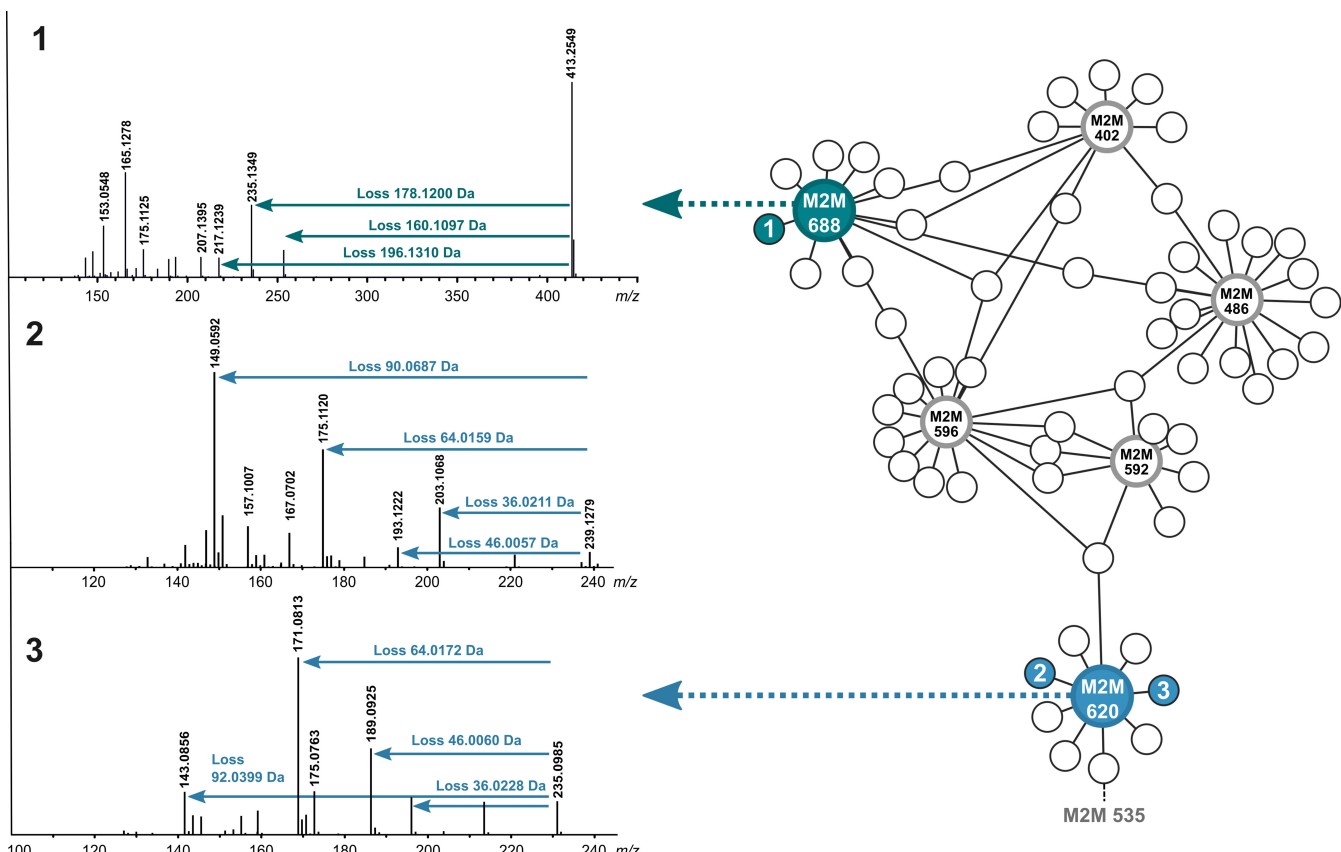

**FIG 5** MS2LDA analysis of the secondary metabolites isolated from the extract of the oatmeal solid culture of *D. caliensis* sp. nov. Mass2Motif 620 and 688 were found to be representative of the caliensolides A (**2**) and B (**3**), and phomol (**1**), respectively. The presence of a node containing Mass2Motifs 592, 596, and 620 agrees with the observed MF containing both types of compounds in the MN.

since one molecule (*m/z* 269.1389 Da) contains, apart from the Mass2Motif 620, two other Mass2Motifs (592 and 596), which are related to the phomol-like derivatives. The Mass2Motif 688 was in the MS/MS spectrum of phomol, along with 11 other different features, of which five also contained other Mass2Motifs such as 402, 486, and 596. The chemical complexity observed for the different phomol derivatives is the reason why the phomol MF represented the biggest cluster in the FBMN (Fig. 5).

In order to understand the identified Mass2Motifs, we investigated the fragmentation patterns and correlated them to the respective structural features. In MS/MS experiments, phomol generated fragment ions at 253.1452, 235.1349, 217.1220, 165.1278, and 153.0548 Da as well as smaller but less intense fragments (Fig. S21). For instance, the 160.1097 Da neutral loss could be interpreted as the loss of the side chain at C–8. The above suggests that the neutral losses associated with the Mass2Motif 688 represent the side chain loss at C–8 and the corresponding losses of the hydroxyl groups in the macrolide. Analogously, MS fragmentation patterns for caliensolides A and B (Fig. S22 and S23) explained by the Mass2Motif 620 corresponded to losses from the lactone ring and the side chain at C–7. The different neutral losses in the fragmentation spectra for both compounds could be attributed to the side chain at C–6, which indeed is the part of the molecule driving the chemical differences between both molecules.

## Biological activities

We traced back the observed activity in the crude extract produced by *D. caliensis* sp. nov. to the phomol (**1**), which showed moderate fungicidal activity against *Mucor hiemalis*, weak activity against *Rhodotorula glutinis* and Gram-positive bacteria. On

the other hand, the new compounds **2** and **3** did not exhibit any activity against the evaluated microorganisms in the serial dilution assay (Table S3). Only compound **1** showed strong cytotoxic activity against the mammalian cell lines tested, while compounds **2** and **3** did not have any activity (Table S4).

## DISCUSSION

Species belonging to the genus *Diaporthe* (Diaporthaceae, Diaporthales) are rather ubiquitous and have often been described as plant pathogens, endophytes, or saprobes in terrestrial host plants, and even causing health problems in humans and other mammals (13, 29). To date, more than 350 secondary metabolites have been reported from this genus (29). Mainly these compounds are classified between polyketides, terpenoids, steroids, macrolides, 10-membered lactones, alkaloids, flavonoids, and fatty acids, for which several important biological properties have been identified (13, 29). Despite the increasing number of discovered metabolites from this genus, most of the producer strains remain with an uncertain taxonomical identification. Taking into account the great success of polyphasic approaches in other ascomycetes such as *Aspergillus* and *Penicillium* (30), or species of the family Hypoxylaceae (31–33), a better understanding of the phylogenetic placement of species of *Diaporthe* will improve further screening programs.

Our conventional chemical screening approach led to the isolation of the bioactive metabolite responsible for the observed antimicrobial activity in the crude extract. Phomol (**1**) was first isolated from an endophytic *Diaporthe* (producer was referred to the outdated name *Phomopsis*) strain isolated from the medicinal plant *Erythrina crista-galli*, for which antifungal, antibacterial, and anti-inflammatory activities were reported (25). This metabolite belongs to the 10-membered lactones, which are well known for their antitumor, anti-inflammatory, antiviral, antibacterial, and other valuable pharmaceutical applications (29). The phomolides are another group of closely related metabolites from *Diaporthe*. For instance, phomolide C exhibited inhibitory effects against the proliferation of human colon adenocarcinoma cells at a concentration of 50 µg/mL (34). Xylarolide and xylarolide A, other 10-membered lactones isolated from endophytic *Diaporthe* spp., have shown interesting cytotoxic properties (35). Furthermore, screening of the endolichenic *Hypoxylon fuscum* has also yielded another 10- membered macrolide named 5,6-epoxyphomol, besides the known phomol (**3**) (26). In this study, phomol (**1**) showed moderate fungicidal activity against *M. hiemalis* and weak activity against *R. glutinis*, which might arise due to the high toxicity of **1** against eukaryotic cells. However, it is not clear whether the activity observed against Gram-positive bacteria arises from the same reason, and, therefore, further investigation on the mode of action of these 10-membered macrolides will be necessary.

Additionally, two previously undescribed secondary metabolites were isolated. Caliensolides A and B (**2** and **3**) belong to the butenolides or butyrolactones, a class of polyketide-lactones with a four-carbon heterocyclic ring structure. Butenolides have been reported to exhibit several biological activities, including anti-inflammatory, cytotoxic, antiviral, antioxidant, antimicrobial, antidiabetic, protein kinase-inhibitory, α-glucosidase-inhibitory, and phytotoxic activities (36). The investigation of secondary metabolites production from *Diaporthe* spp. isolated from grapevine plants led to the discovery of phomopsolidones A and B (37). These compounds displayed phytotoxic and antibacterial activities; however, when compared to their pyranone cometabolite, phomopsolide B, they presented weaker bioactivity. This fact has been previously reported in the case of the observed herbicidal activity for phomolactone A and its butenolide cometabolites (38). Herein, compounds **2** and **3** did not show any activity against the tested microorganisms or cytotoxic activity against the two tested cell lines.

The metabolomic investigation of *D. caliensis* sp. nov. illustrated among others the richness of the produced metabolites (99.5% unknown features) as well as the limitations of the current tools to classify the observed metabolites. The phenomenon of individual spectra containing multiple substructure annotations showed the incapacity of

the molecular networking approach to cluster separately chemically distinct metabolites. This also emphasizes the value of having reference spectra for comparison and validation of automated annotations. In fact, the evaluation of FBMN topologies without a comprehensive chemical knowledge of the studied samples yields results that are misleading. For instance, the presence of multiple motifs within the same spectra directly influences the power and resolution of the cosine similarity as a parameter to delimitate different compound classes based on spectral similarity, as shown herein for the phomol and the caliensolides. However, further inspection of the same data set using the MS2LDA workflow provided a better understanding of this phenomenon and allowed the identification of Mass2Motifs driving chemical similarity among the different molecules in the extract produced by *D. caliensis* sp. nov. The present study exemplifies the taxonomical diversity that can be obtained in countries of the Neotropics, such as Colombia, for well-studied genera. *Diaporthe caliensis* sp. nov. represents the 10th native species of this genus in Colombia, which highlights the value of upcoming campaigns to explore the fungal biodiversity of the country. What is more, it is to be expected that such taxonomical diversity can result in new sources to obtain novel natural products with potential beneficial applications.

## MATERIALS AND METHODS

### Fungal isolation

Fresh and healthy leaves from young trees of *Otoba gracilipes* were collected during the dry season (November 2019) in the Natural Reserve "La Carolina" (3°24'10.662"N, 76°36'52.774"W), Cali, Valle del Cauca, Colombia at 1,600 m.a.s.l. The plant material was surface-disinfected by washing thoroughly in sterile demineralized water, followed by 70% ethanol for 1–2 min, and 3% sodium hypochlorite (NaClO) for 15 min. Small sections of plant tissue were then placed on PDA (Merck, Darmstadt, Germany; pH = 6), supplemented with clindamycin (0.2%, vol/vol), and incubated at 29°C. After sufficient fungal growth, the colonies were examined, and hyphal tips were transferred to PDA using a sterile needle and incubated at 29°C (12).

Herbarium and ex-type material of the new species are maintained at the Colección de microorganismos-Escuela de Microbiología, UdeA (HUA; CM-EM-UDEA), Medellin, Colombia. Isotype strains were also deposited at the collection of the Westerdijk Fungal Biodiversity Institute (CBS), Utrecht, the Netherlands, and in the fungarium of the Helmholtz Centre for Infection Research (STMA), Braunschweig, Germany.

### Phenotypic characterization

The isolate was grown for 15 days on MEA (HiMedia, Mumbai, India), OA (Sigma-Aldrich, St. Louis, Missouri, USA), and PDA at 21°C in darkness for the cultural characterization (39). Color notations in parentheses are taken from the color chart of The Royal Horticultural Society London (40). The fungus was grown in 2% tap water agar supplemented with sterile pine needles (PNA; 41) to induce sporulation.

### Molecular study

The extraction of DNA of the fungus and the amplification and sequencing of the internal transcribed spacers and intervening 5.8S nrDNA (ITS), and fragments of the calmodulin (*cal*), histone H3 (*his3*), partial elongation factor 1-alpha (*tef1*), and β-tubulin (*tub2*) genes were performed as in Matio Kemkuignou et al. (42).

A BLAST search was conducted to estimate its affinities to other described species available in GenBank, suggesting that it could represent a novel taxon according to the nucleotide similarity to the other *Diaporthe* spp. and revealing a close relationship with a group of species we had recently examined in a previous study (42). The ML and Bayesian inference analyses, comprising the five loci of the same species included in this previous study (Table S1), were both performed as described by Harms et al. (43).

## Fermentation, extraction, and isolation

The fungus was grown in yeast malt (YM) agar at 23°C. Later, the colonies were cut into small pieces using a cork borer (1 cm × 1 cm) and eight pieces were placed into a 500 mL Erlenmeyer flask containing 200 mL of yeast malt extract broth (YM broth; malt extract 10 g/L, yeast extract 4 g/L, D‑glucose 4 g/L, pH 6.3 before autoclaving), which was incubated at 23°C under shaking conditions (140 rpm) in the darkness. After 7 days, 6 mL of this seed culture was transferred to each of the 12 conical flasks of 500 mL containing solid oat medium [OFT; oat 28 g, as well as 80 mL of base liquid (yeast extract 1 g/L, di‑sodium tartrate di‑hydrate 0.5 g/L, $KH_2PO_4$ 0.5 g/L per flask)]. The OFT cultures were incubated for 15 days at 23°C in darkness. To extract the secondary metabolites, the mycelia were covered with acetone, and sonicated in an ultrasonic bath for 30 min at 40°C. The acetone extract was separated from the mycelia by filtration throughout a cellulose filter paper (MN 615 1/4 Ø 185 mm, Macherey‑Nagel, Düren, Germany), and the remaining mycelia were sonicated and extracted again. Both extracts were combined, and the acetone was evaporated to yield an aqueous residue *in vacuo* at 40°C. The resulting aqueous phase was extracted twice with an equal amount of ethyl acetate in a separatory funnel. The ethyl acetate fraction was evaporated to dryness *in vacuo* at 40°C. Afterward, the ethyl acetate extract was dissolved in methanol and partitioned with an equal amount of heptane in a separatory funnel. This later step was repeated with the obtained methanol phase, which was then evaporated to dryness *in vacuo* at 40°C. Both extracts were combined and dried *in vacuo* at 40°C to afford 1153 mg of crude extract.

The crude extract was pre-fractionated using flash chromatography (Grace Reveleris, Columbia, MD, USA) {silica cartridge 40 g, the mobile phase consisted of A [DCM], B [acetone], and C [(DCM/acetone 8:2):MeOH], gradient: 100% A for 5 min, increasing to 100% B in 20 min, followed by increasing to 100% solvent mixture C in 15 min and holding at 100% solvent C in 5 min}. Five fractions (F1–F5) were collected. Fraction F1 was found to contain almost exclusively fatty acids and subsequently discharged.

Fraction F2 (280 mg) was purified by preparative reversed phase-HPLC [Synergi Polar-RP column (250 × 50 mm, 10 µm, Phenomenex, Torrance, CA, USA), the mobile phase consisted of A ($H_2O$ + 0.1% formic acid) and B (ACN + 0.1% formic acid), flow rate 45 mL/min and UV detection at 210, 240, and 300 nm, gradient: from 5% to 32% B in 15 min, from 32% to 36% in 40 min, from 36% to 100% in 25 min, and 100% B isocratic for 5 min] to afford caliensolide A (**2**) (3.29 mg, tR = 41–43 min), caliensolide B (**3**) (2.39 mg, tR = 56–57 min), and phomol (**1**) (11 mg, tR = 71.5–72.5 min).

## Untargeted metabolomics analyses

Each sample was analyzed using the instrumental settings and conditions reported previously (44). Raw data were pre-processed with MetaboScape 2022 (Bruker Daltonics, Bremen, Germany) in the retention time range of 0.5–20 min, and the obtained features were dereplicated based on their accurate molecular weight and MS/MS spectra against the compounds reported under *Diaporthe* or *Phomopsis* in the Natural Product Atlas (NP Atlas) database (22). For this purpose, MetaboScape performed automatic *in silico* MS/MS matching based on the InChI-encoded structures using the MetFrag algorithm in the absence of MS/MS reference data (21).

Molecular networks were created with the FBMN (23) workflow on the GNPS platform (24) using the pre-processed feature table from MetaboScape as described by Charria-Girón et al. (44). The spectra in the network were then searched against GNPS spectral libraries (24, 45). The molecular networks were visualized using Cytoscape software (46). Additionally, in-depth exploration of substructural correlations was performed using the MS2LDA workflow (28). Also, MolNetEnhancer was used to propagate the annotation of chemical classes to the full subnetwork (47). Detailed spectral data and GNPS result links are provided in the supplemental material.

## Spectral data

Optical rotations were recorded employing an MCP 150 circular polarimeter (Anton Paar, Seelze, Germany) at 20°C. UV/Vis spectra were recorded with a UV‑2450 spectrophotometer (Shimadzu, Kyoto, Japan). Spectral data were measured in MeOH (Uvasol, Merck, Darmstadt, Germany) for all compounds. The 1D‑ and 2D‑nuclear magnetic resonance (NMR) spectra were recorded with an Avance III 700 spectrometer with a 5 mm TCI cryoprobe ($^1$H NMR: 700 MHz, $^{13}$C: 175 MHz, Bruker, Billerica, MA, USA) and an Avance III 500 spectrometer ($^1$H NMR: 500 MHz, $^{13}$C: 125 MHz, Bruker, Billerica, MA, USA). The chemical shifts δ were referenced to the solvents DMSO‑$d_6$ ($^1$H, δ = 2.50; $^{13}$C, δ = 39.51).

### Phomol (1)

Colorless oil; $^1$H NMR (700 MHz, pyridine-$d_5$): $\delta_H$ 7.38 (br d, $J$ = 4.3 Hz, 7–OH), 6.86 (dq, $J$ = 10.1, 1.3 Hz, 3′–H), 6.56 (ddd, $J$ = 15.6, 9.5, 2.1 Hz, 5–H), 6.21 (dd, $J$ = 15.6, 2.1 Hz, 6–H), 6.11 (ddd, $J$ = 9.9, 9.1, 3.0 Hz, 9–H), 5.39 (dd, $J$ = 9.9, 2.2 Hz, 8–H), 5.13 (ddd, $J$ = 4.3, 2.2, 2.1 Hz, 7–H), 4.65 (ddd, $J$ = 11.3, 8.5, 3.0 Hz, 3–H), 4.56 (dd, $J$ = 9.5, 8.5 Hz, 4–H), 3.09 (dd, $J$ = 13.2, 3.0 Hz, 2–H$_a$), 2.82 (dd, $J$ = 13.2, 11.3 Hz, 2–H$_b$), 2.33 (m, 4′–H), 1.94 (d, $J$ = 1.1 Hz, 7′–H), 1.84 (m, 10–H$_a$), 1.68 (m, 10–H$_b$), 1.44 (m, 11–H), 1.27 (m, 12–H$_a$), 1.25 (m, 5–H$_a$), 1.194 (m, 13–H$_2$), 1.190 (m, 12–H$_b$), 1.16 (m, 5–H$_b$), 0.89 (d, $J$ = 6.7 Hz, 8′–H$_3$), 0.78 (t, $J$ = 7.0 Hz, 14–H$_3$), 0.78 (t, $J$ = 7.4 Hz, 6′–H$_3$) ppm; $^{13}$C NMR (175 MHz, pyridine-$d_5$): $\delta_C$ 172.3 (C, C–1), 167.9 (C, C–1′), 149.5 (CH, C–3′), 133.6 (CH, C–6), 127.7 (CH, C–5), 127.4 (C, C–2′), 80.9 (CH, C–4), 76.0 (CH, C–8), 74.1 (CH, C–3), 70.2 (CH, C–7), 69.1 (CH, C–9), 42.3 (CH$_2$, C–2), 35.4 (CH, C–4′), 32.2 (CH$_2$, C–12), 32.1 (CH$_2$, C–10), 30.0 (CH$_2$, C–5′), 24.7 (CH$_2$, C–11), 23.0 (CH$_2$, C–13), 20.0 (CH$_3$, C–8′), 14.5 (CH$_3$, C–14), 13.2 (CH$_3$, C–7′), 12.3 (CH$_3$, C–6′) ppm; high-resolution electrospray ionization mass spectrometry: $m/z$ 435.2352 [M + Na]$^+$ (calculated for $C_{22}H_{36}O_7Na^+$: 435.2353 Da) and 413.2538 [M + H]$^+$ (calculated for $C_{22}H_{37}O_7^+$: 413.2534 Da).

### Caliensolide A (2)

Orange oil; [α]$_D^{20}$ 28 (c 0.001, MeOH); UV (MeOH) λmax (log ε) 203 (4.02); $^1$H NMR (500 MHz, DMSO-$d_6$): $\delta_H$ 7.28 (dd, $J$ = 15.6, 10.9 Hz, 9–H), 6.51 (d, $J$ = 15.6 Hz, 8–H), 6.47 (ddd, $J$ = 17.1, 10.9, 10.4 Hz, 10–H), 5.40 (d, $J$ = 17.1 Hz, 11–H$_a$), 5.24 (br d, $J$ = 10.4 Hz, 11–H$_b$), 4.92 (d, $J$ = 18.6 Hz, 13–H$_a$), 4.86 (d, $J$ = 18.6 Hz, 13–H$_b$), 4.49 (d, $J$ = 6.6 Hz, 5–H), 3.51 (m, 4–H), 1.54 (m, 3–H$_a$), 1.48 (m, 2–H$_a$), 1.29 (m, 2–H$_b$), 1.25 (m, 3–H$_b$), 0.87 (t, $J$ = 7.2 Hz, 1–H$_3$) ppm; $^{13}$C NMR (125 MHz, DMSO-$d_6$): $\delta_C$ 172.2 (C, C–12), 163.9 (C, C–6), 137.4 (CH, C–10), 133.2 (CH, C–9), 121.7 (C, C–7), 121.4 (CH, C–8), 119.8 (CH$_2$, C–11), 72.4 (CH, C–4), 69.9 (CH, C–5), 69.8 (CH$_2$, C–13), 35.2 (CH$_2$, C–3), 18.3 (CH$_2$, C–2), 14.0 (CH$_3$, C–1) ppm; ESI‑MS: $m/z$ 236.82 [M − H]$^-$, 239.02 [M + H]$^+$, and 261.03 [M + Na]$^+$; HRESI-MS: $m/z$ 239.1283 [M + H]$^+$ (calculated for $C_{13}H_{19}O_4^+$: 239.1278 Da).

### Caliensolide B (3)

White powder; UV (MeOH) λmax (log ε) 263 (3.35), (log ε) 202.5 (3.06); $^1$H NMR (500 MHz, DMSO-$d_6$): $\delta_H$ 7.30 (dd, $J$ = 15.6, 10.9 Hz, 9–H), 6.80 (d, $J$ = 16.0 Hz, 5–H), 6.66 (d, $J$ = 15.6 Hz, 8–H), 6.49 (ddd, $J$ = 17.1, 10.9, 10.4 Hz, 10–H), 6.27 (dt, $J$ = 16.0, 6.4 Hz, 4–H), 5.43 (d, $J$ = 17.1 Hz, 11–H$_a$), 5.24 (d, $J$ = 10.4 Hz, 11–H$_b$), 5.02 (s, 13–H$_2$), 2.40–2.50 (m, 2–H$_2$, 3–H$_2$) ppm; $\delta_C$ 173.6 (C, C–1), 172.2 (C, C–12), 154.3 (C, C–6), 140.1 (CH, C–4), 137.4 (CH, C–10), 133.5 (CH, C–9), 120.8 (CH, C–8), 120.4 (CH, C–5), 120.1 (CH$_2$, C–11), 118.9 (CH, C–7), 68.7 (CH$_2$, C–13), 32.4 (CH$_2$, C–2), 28.2 (CH$_2$, C–3) ppm; ESI‑MS: $m/z$ 232.79 [M − H]$^-$, 235.00 [M + H]$^+$, and 257.01 [M + Na]$^+$; HRESI-MS: $m/z$ 235.0970 [M + H]$^+$ (calculated for $C_{13}H_{15}O_4^+$: 235.0965 Da).

## Derivatization of phomol (1) with MTPA

Phomol (**1**) was dissolved in pyridine-$d_5$ (50 µL), transferred into a 250 µL glass vial, and then (R)-(−)-α-methoxy-α-(trifluoromethyl) phenylacetyl chloride (4 µL) was added. The mixture was incubated for 2 h at room temperature before being transferred to an NMR tube (600 µL) and diluted to a final volume of 350 µL for the measurement of ¹H, COSY, TOCSY, HSQC, and HMBC NMR spectra. This resulted in the (S)-MTPA ester derivative: ¹H NMR (700 MHz, pyridine-$d_5$): similar to **1**, but $\delta_H$ 6.84 (br d, J = 10.1 Hz, 3′–H), 6.70 (br d, J = 15.5 Hz, 6–H), 6.56 (ddd, J = 15.6, 9.5, 2.1 Hz, 5–H), 6.24 (m, 4–H), 6.10 (m, 3–H), 6.08 (m, 9–H), 5.37 (dd, J = 9.9, 2.2 Hz, 8–H), 5.15 (m, 7–H), 3.30 (dd, J = 13.2, 3.0 Hz, 2–H$_a$), 3.00 (dd, J = 13.2, 11.3 Hz, 2–H$_b$), 2.31 (m, 4′–H), 1.91 (d, J = 1.1 Hz, 7′–H), 1.83 (m, 10–H$_a$), 1.66 (m, 10–H$_b$), 1.39 (m, 11–H), 1.27 (m, 12–H$_a$), 1.23 (m, 5–H$_2$), 1.21 (m, 13–H$_2$), 1.20 (m, 12–H$_b$), 0.88 (d, J = 6.7 Hz, 8′–H$_3$), 0.80 (m, 14–H$_3$), and 0.72 (t, J = 7.4 Hz, 6′–H$_3$) ppm.

The (R)-MTPA ester derivative was obtained analogously with (S)-(+)-α-methoxy-α-(trifluoromethyl) phenylacetyl chloride (4 µL): ¹H NMR (700 MHz, pyridine-$d_5$): similar to **1**, but $\delta_H$ 6.83 (br d, J = 10.1 Hz, 3′–H), 6.68 (br d, J = 15.5 Hz, 6–H), 6.19 (ddd, J = 15.6, 9.5, 2.1 Hz, 5–H), 6.14 (m, 4–H), 6.07 (m, 3–H), 6.02 (m, 9–H), 5.34 (dd, J = 9.9, 2.2 Hz, 8–H), 5.16 (m, 7–H), 3.20 (dd, J = 13.2, 3.0 Hz, 2–H$_a$), 2.82 (dd, J = 13.2, 11.3 Hz, 2–H$_b$), 2.31 (m, 4′–H), 1.91 (d, J = 1.1 Hz, 7′–H), 1.81 (m, 10–H$_a$), 1.63 (m, 10–H$_b$), 1.36 (m, 11–H), 1.23 (m, 12–H$_a$), 1.23 (m, 5–H$_a$), 1.19 (m, 13–H$_2$), 1.18 (m, 12–H$_b$), 1.16 (m, 5–H$_b$), 0.87 (d, J = 6.7 Hz, 8′–H$_3$), 0.78 (d, J = 7.0 Hz, 14–H$_3$), and 0.71 (t, J = 7.4 Hz, 6′–H$_3$) ppm.

## Ozonolysis of phomol (1) and GC/MS analysis

Phomol (**1**) was ozonized (1.0 mL $CH_2Cl_2$, 20 min, −78°C) with oxidative workup (five drops of 30% $H_2O_2$). The excess reagent was evaporated under a constant stream of $N_2$ gas. All samples were analyzed using a chiral GC-MS under identical conditions on a Trace GC-Ultra gas chromatograph coupled to a Thermo Scientific ITQ900 mass spectrometer (both from Thermo Scientific, Dreieich, Germany). A Hydrodex β-PM column with 50 m × 0.25 mm ID was used for separation (Macherey-Nagel, Düren, Germany). The following thermal gradient was used: initial oven temperature of 70°C was held for 2.5 min, then the temperature was increased to 100°C at a rate of 1.5°C/min, followed by a temperature increase to 220°C at a rate of 4°C/min and held for 5 min. The spectral data were collected from 35 to 650 Da in positive ion mode, and the transferring line and ion source temperatures were both set to 220°C. Data acquisition and processing were done with XCalibur Software. Co-injection of the (S)-2-methyl butyric acid standard with the fragment from **1** gave one peak at 30.02 min, while co-injection of the (R)-2-methyl butyric acid standard with the fragment from **1** gave two peaks at 30.03 min (phomol) and 30.35 min (R). Thus, the absolute configuration at C–4′ was assigned as S.

## Derivatization of caliensolide A (2) with MTPA

The Mosher's method experiment on **2** was analogously carried out as described for **1**. The (S)-MTPA ester derivative was obtained with (R)-(+)-α-methoxy-α-(trifluoromethyl) phenylacetyl chloride (4 µL): ¹H NMR (700 MHz, pyridine-$d_5$): similar to **2**, but $\delta_H$ 7.94 (m, 9–H), 6.96 (br d, J = 15.5 Hz, 8–H), 6.90 (m, 5–H), 6.51 (m, 10–H), 6.03 (dt, J = 8.3, 4.1 Hz, 4–H), 5.45 (br d, J = 16.6 Hz, 11–H$_a$), 5.26 (br d, J = 9.9 Hz, 11–H$_b$), 4.76 (d, J = 18.8 Hz, 13–H$_a$), 4.07 (d, J = 18.8 Hz, 13–H$_b$), 1.80 (m, 3–H$_2$), 1.48 (m, 2–H$_2$), and 0.86 (t, J = 7.3 Hz, 1–H$_3$).

In contrast, the (R)-MTPA ester derivative was obtained with (S)-(+)-α-methoxy-α-(trifluoromethyl) phenylacetyl chloride (4 µL): ¹H NMR (700 MHz, pyridine-$d_5$): similar to **2**, but $\delta_H$ 7.96 (m, 9–H), 6.98 (br d, J = 9.9 Hz, 8–H), 6.91 (m, 5–H), 6.52 (m, 10–H), 5.89 (m, 4–H), 5.45 (br d, J = 16.6 Hz, 11–H$_a$), 5.27 (br d, J = 10.5 Hz, 11–H$_b$), 5.12 (d, J = 18.8 Hz, 13–H$_a$), 4.94 (d, J = 18.8 Hz, 13–H$_b$), 1.60 (m, 3–H$_2$), 1.27 (m, 2–H$_2$), and 0.87 (m, 1–H$_3$).

## Antimicrobial and cytotoxic activity assays

The antimicrobial and cytotoxic assays were performed according to the methods reported previously (43, 48).

## ACKNOWLEDGMENTS

We are grateful to Wera Collisi for conducting the antimicrobial and cytotoxicity assays, Christel Kakoschke and Kirsten Harmrolfs for recording NMR data, and Lena Schweizer for providing the sequencing data. The authors are also grateful to Karoline Jerye, Mathias Göhl, Fabio Brescia, Martin Rühl, Sebastian Pfütze, and Jan-Peer Wennrich for their expert advisory assistance.

The genetic resource access contract RGE244-44#190 was provided by the Autoridad Nacional de Licencias Ambientales (ANLA) from the Ministerio de Ambiente y Desarrollo Sostenible in Colombia.

Esteban Charria-Girón was supported by a grant (No: 22028) from the Ministerio de Ciencias, Tecnología e Innovación, Colombia, funded to N.H.C., and the HZI POF IV Cooperativity and Creativity Project Call funded to F.S. Y.M.-F. was supported by the Deutsche Forschungsgemeinschaft (DFG)—Project-ID 490821847. A.M.V.-P. was supported by Primer Proyecto, project 2020–33675.

## AUTHOR AFFILIATIONS

[1]Department Microbial Drugs, Helmholtz Centre for Infection Research (HZI), German Centre for Infection Research (DZIF), Partner Site Hannover-Braunschweig, Braunschweig, Germany

[2]Institute of Microbiology, Technische Universität Braunschweig, Braunschweig, Germany

[3]Departamento de Ciencias biológicas, Facultad de Ingeniería, Diseño y Ciencias Aplicadas, Bioprocesos y Biotecnología, Universidad Icesi, Cali, Colombia

[4]Department Chemical Biology, Helmholtz Centre for Infection Research GmbH (HZI), Braunschweig, Germany

[5]Grupo de Microbiología Ambiental y Grupo BioMicro, Escuela de Microbiología, Universidad de Antioquia UdeA, Medellín, Colombia

[6]Asociación Colombiana de Micología, ASCOLMIC, Bogotá, Colombia

[7]Centro BioInc, Universidad Icesi, Cali, Colombia

## AUTHOR ORCIDs

Esteban Charria-Girón ⓘ http://orcid.org/0000-0002-3823-1416
Yasmina Marin-Felix ⓘ http://orcid.org/0000-0001-8045-4798
Raimo Franke ⓘ http://orcid.org/0000-0002-5407-5521
Mark Brönstrup ⓘ http://orcid.org/0000-0002-8971-7045
Nelson H. Caicedo ⓘ http://orcid.org/0000-0002-2133-7994
Frank Surup ⓘ http://orcid.org/0000-0001-5234-8525

## FUNDING

| Funder | Grant(s) | Author(s) |
| --- | --- | --- |
| Ministerio de Ciencia, Tecnología, Conocimiento e Innovación (CTCI) | 22028 | Nelson H. Caicedo |
| Helmholtz-Fonds (Helmholtz-Fonds e.V.) | CCC grant | Frank Surup |
| Deutsche Forschungsgemeinschaft (DFG) | 490821847 | Yasmina Marin-Felix |
| Primer Proyecto, project 2020 | 33675 | Aida M. Vasco-Palacios |

## AUTHOR CONTRIBUTIONS

Esteban Charria-Girón, Formal analysis, Writing – original draft | Yasmina Marin-Felix, Data curation, Formal analysis, Methodology, Project administration, Supervision, Writing – original draft | Ulrike Beutling, Data curation, Formal analysis | Raimo Franke, Data curation, Formal analysis, Software, Writing – review and editing | Mark Brönstrup, Resources, Supervision, Writing – review and editing | Aida M. Vasco-Palacios, Investigation, Resources | Nelson H. Caicedo, Conceptualization, Formal analysis, Resources | Frank Surup, Formal analysis, Funding acquisition, Investigation, Methodology, Project administration, Supervision, Writing – original draft, Writing – review and editing

## ADDITIONAL FILES

The following material is available online.

### Supplemental Material

**Supplemental material (Spectrum02743-23-S0001.docx).** Fig. S1 to S23 and Tables S1 to S5.

### Open Peer Review

**PEER REVIEW HISTORY (review-history.pdf).** An accounting of the reviewer comments and feedback.

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
