## [Reviewer comments · Microbiology Spectrum]

Microbiology Spectrum

Metabolomics insights into the polyketide-lactones produced by *Diaporthe caliensis* sp. nov., an endophyte of the medicinal plant *Otoba gracilipes*

Esteban Charria-Girón, Yasmina Marin-Felix, Ulrike Beutling, Raimo Franke, Mark Brönstrup, Aida Vasco-Palacios, Nelson Caicedo, and Frank Surup

Corresponding Author(s): Frank Surup, Helmholtz-Zentrum für Infektionsforschung GmbH

Review Timeline:

Submission Date:	July 4, 2023
Editorial Decision:	August 31, 2023
Revision Received:	September 25, 2023
Accepted:	October 1, 2023

Editor: Sacha Pidot

Reviewer(s): Disclosure of reviewer identity is with reference to reviewer comments included in decision letter(s). The following individuals involved in review of your submission have agreed to reveal their identity: Alison Felipe Alencar Chaves (Reviewer #1)

Transaction Report:

DOI: <https://doi.org/10.1128/spectrum.02743-23>

August 31, 2023

Dr. Frank Surup
Helmholtz-Zentrum für Infektionsforschung GmbH
Mikrobielle Wirkstoffe
Inhoffenstraße 7
Braunschweig 38124
Germany

Re: Spectrum02743-23 (Metabolomics insights into the polyketide-lactones produced by *Diaporthe caliensis* sp. nov., an endophyte of the medicinal plant *Otoba gracilipes*)

Dear Dr. Frank Surup:

Link Not Available

Sincerely,

Sacha Pidot

Journals Department
Reviewer comments:

Reviewer #1 (Comments for the Author):

The authors produced a valuable study evaluating the secondary metabolites from a new species of *Diaporthe* fungus, classified as *D. caliense* sp. nov.. and found two new metabolites named *caliensolides* A and B. The authors applied an interesting pipeline for the data analysis which highlight the solving problems abilities of the group. The study could extract the information even with the annotation problems inherent to fungi metabolome databases. They solved the *caliensolide* A structure showing the determination to find the answers. Yet, they surpassed the limitation of the FBMN approach adjusting the cosine score and added value using MS2LDA. That said, this reviewer has some minor question and observations:

1. In the Figure S19, where the authors show the cosine score adjustment, maybe it is worth to add the caveat about the limitation of this molecular network topology analysis without a robust support of the structural chemistry data.
2. One suggestion for future work trying to induce the sporulation would be to try the Microcultivation technique.
3. In the table S2 and S3, the concentration must to be consistent. This reviewer recommends μM units, as presented in table S3.
4. The inconsistency in the units for compounds concentration avoid to see the actual cytotoxicity effect of the phomol. Please note that, if the information provided in the table S2 are correct, then the statement in the biological activity session "compound 1 showed weak cytotoxic activity against the mammalian cell lines tested" and in the discussion session "weak cytotoxic effects against the human endocervical adenocarcinoma and mouse fibroblasts evaluated cell lines" do not holds. As presented by the reference number 34, phomolide C had activity against human tumor cell at 50 $\mu\text{g}/\text{mL}$ close to the concentrations found in the present manuscript ($\sim 66 \mu\text{g}/\text{mL}$) against the microorganisms. In molar concentration this 66 $\mu\text{g}/\text{mL}$ would be equivalent to approximately 161 μM , which is way higher than the IC50 for the fibroblast cell line. The biological activity is due to high toxicity of phomol or specific activity against the microorganisms? This point needs a solid clarification.

Reviewer #2 (Comments for the Author):

Charria-Girón et al. successfully isolated a novel endophyte named *Diaporthe caliensis*. They conducted secondary metabolomics to characterize the molecular features of the culture extract, which were subsequently analyzed using molecular networking techniques. In their study, they not only isolated a known compound, phomol (1), but also identified two new structurally distinct molecules named caliensolides A (2) and B (3). Furthermore, they determined the absolute stereochemistry of secondary alcohols in compounds 1 and 2 using Mosher's method.

While I find that some aspects of the manuscript might not align perfectly with the scope of the journal, the sections focusing on the isolation and elucidation of the compound structures were executed excellently and hold significant merit for potential publication.

1. Table 1 can be moved to the Supplementary Information (SI). Consequently, citation references within the table should also be transferred to the SI.
2. Figure 1 is referenced but not included in the manuscript.
3. In Table S4, the current format is not suitable. Reevaluate the presentation style to include appropriate protein alignment formatting. Additionally, ensure that the table numbers follow the order of appearance.
4. The excessive use of abbreviations can hinder readability. There are repeated instances of abbreviation use. When an abbreviation is introduced for the first time, spell it out completely.
5. Lines 96-109 are challenging to comprehend. Consider revising the presentation style to enhance clarity.
6. Lines 110-123: Incorporating a photographic image of the culture would facilitate the understanding of the description.
7. Lines 143-152: Including figure numbers would aid in understanding the description.

Minor Corrections:

Line 243: *Mucor hiemalis*.

Line 244: *Rhodotorula glutinis*.

Line 273: phomol (3) should be corrected to phomol (1).

Line 287: Change "1 and 2" to "2 and 3".

Staff Comments:

Preparing Revision Guidelines

Please return the manuscript within 60 days; if you cannot complete the modification within this time period, please contact me. If you do not wish to modify the manuscript and prefer to submit it to another journal, please notify me of your decision immediately so that the manuscript may be formally withdrawn from consideration by Microbiology Spectrum.

Review Spectrum 311399

The authors produced a valuable study evaluating the secondary metabolites from a new species of *Diaporthe* fungus, classified as *D. caliense* sp. nov.. and found two new metabolites named caliensolides A and B. The authors applied an interesting pipeline for the data analysis which highlight the solving problems abilities of the group. The study could extract the information even with the annotation problems inherent to fungi metabolome databases. They solved the caliensolide A structure showing the determination to find the answers. Yet, they surpassed the limitation of the FBMN approach adjusting the cosine score and added value using MS2LDA. That said, this reviewer has some minor question and observations:

1. In the Figure S19, where the authors show the cosine score adjustment, maybe it is worth to add the caveat about the limitation of this molecular network topology analysis without a robust support of the structural chemistry data.
2. One suggestion for future work trying to induce the sporulation would be to try the Microcultivation technique.
3. In the table S2 and S3, the concentration must to be consistent. This reviewer recommends μM units, as presented in table S3.
4. The inconsistency in the units for compounds concentration avoid to see the actual cytotoxicity effect of the phomol. Please note that, if the information provided in the table S2 are correct, then the statement in the biological activity session "compound 1 showed weak cytotoxic activity against the mammalian cell lines tested" and in the discussion session "weak cytotoxic effects against the human endocervical adenocarcinoma and mouse fibroblasts evaluated cell lines" do not holds. As presented by the reference number 34, phomolide C had activity against human tumor cell at $50 \mu\text{g}/\text{mL}$ close to the concentrations found in the present manuscript ($\sim 66 \mu\text{g}/\text{mL}$) against the microorganisms. In molar concentration this $66 \mu\text{g}/\text{mL}$ would be equivalent to approximately $161 \mu\text{M}$, which is way higher than the IC_{50} for the fibroblast cell line. The biological activity is due to high toxicity of phomol or specific activity against the microorganisms? This point needs a solid clarification.

Rebuttal letter (spectrum02746-23)

Reviewer 1:

The authors produced a valuable study evaluating the secondary metabolites from a new species of Diaporthe fungus, classified as *D. caliense* sp. nov. and found two new metabolites named caliensolides A and B. The authors applied an interesting pipeline for the data analysis which highlights the solving problems abilities of the group. The study could extract the information even with the annotation problems inherent to fungi metabolome databases. They solved the caliensolide A structure showing the determination to find the answers. Yet, they surpassed the limitation of the FBMN approach adjusting the cosine score and added value using MS2LDA. That said, this reviewer has some minor question and observations:

1. In the Figure S19, where the authors show the cosine score adjustment, maybe it is worth to add the caveat about the limitation of this molecular network topology analysis without a robust support of the structural chemistry data.

Thanks a lot for the suggestions, we have included in lines 292 and 293 a statement about this fact.

2. One suggestion for future work trying to induce the sporulation would be to try the Microcultivation technique.

So far, we are not familiar with this technique. However, the authors are grateful to reviewer #1 for his remark, and will try this method in future studies.

3. In the table S2 and S3, the concentration must to be consistent. This reviewer recommends μM units, as presented in table S3.

Table S2 has been modified accordingly.

4. The inconsistency in the units for compounds concentration avoid to see the actual cytotoxicity effect of the phomol. Please note that, if the information provided in the table S2 are correct, then the statement in the biological activity session "compound 1 showed weak cytotoxic activity against the mammalian cell lines tested" and in the discussion session "weak cytotoxic effects against the human endocervical adenocarcinoma and mouse fibroblasts evaluated cell lines" do not hold. As presented by the reference number 34, phomolide C had activity against human tumor cell at $50 \mu\text{g}/\text{mL}$ close to the concentrations found in the present manuscript ($\sim 66 \mu\text{g}/\text{mL}$) against the microorganisms. In molar concentration this $66 \mu\text{g}/\text{mL}$ would be equivalent to approximately $161 \mu\text{M}$, which is way higher than the IC_{50} for the fibroblast cell line. The biological activity is due to high toxicity of phomol or specific activity against the microorganisms? This point needs a solid clarification.

Thanks for pointing out this issue! Indeed the MICs for phomol against the different microorganisms are at least 4 times higher than the obtained IC_{50} s. We have changed the results accordingly and included an additional sentence in the discussion indicating the consequences of these findings.

Reviewer #2:

Charria-Girón et al. successfully isolated a novel endophyte named *Diaporthe caliensis*. They conducted secondary metabolomics to characterize the molecular features of the culture extract, which were subsequently analyzed using molecular networking techniques. In their study, they not only isolated a known compound, phomol (1), but also identified two new structurally distinct molecules named caliensolides A (2) and B (3). Furthermore, they determined the absolute stereochemistry of secondary alcohols in compounds 1 and 2 using Mosher's method.

While I find that some aspects of the manuscript might not align perfectly with the scope of the journal, the sections focusing on the isolation and elucidation of the compound structures were executed excellently and hold significant merit for potential publication.

1. Table 1 can be moved to the Supplementary Information (SI). Consequently, citation references within the table should also be transferred to the SI.

The table 1 was moved into the SI as suggested, same as the corresponding references.

2. Figure 1 is referenced but not included in the manuscript.

Figure 1 is referenced in the text in the line 81 and the consensus ML tree is included in the manuscript at the end of the text (line 855).

3. In Table S4, the current format is not suitable. Reevaluate the presentation style to include appropriate protein alignment formatting. Additionally, ensure that the table numbers follow the order of appearance.

Alignments are always deposit or provided in this fasta format. In fact, it is the only format compatible with all kind of programs used in the phylogenetic analysis. Therefore, the authors consider that no changes are required. Moreover, the phylogenetic analysis is based on sequence data, so it cannot be changed to a protein alignment formatting.

4. The excessive use of abbreviations can hinder readability. There are repeated instances of abbreviation use. When an abbreviation is introduced for the first time, spell it out completely.

We have made the corresponding changes in the revised version

5. Lines 96-109 are challenging to comprehend. Consider revising the presentation style to enhance clarity.

It is the established formatting to introduce a new species based on molecular data. Any other style could invalidate the new species based on the nomenclatural code.

6. Lines 110-123: Incorporating a photographic image of the culture would facilitate the understanding of the description.

Pictures of the colonies are provided in the new manuscript SI.

7. Lines 143-152: Including figure numbers would aid in understanding the description.

Figure numbers have been included in these paragraphs accordingly.

Minor Corrections:

Line 243: *Mucor hiemalis*.

Line 244: *Rhodotorula glutinis*.

Line 273: phomol (3) should be corrected to phomol (1).

Line 287: Change "1 and 2" to "2 and 3".

All minor changes have been modified accordingly.

September 27, 2023

Dr. Frank Surup
Helmholtz-Zentrum für Infektionsforschung GmbH
Mikrobielle Wirkstoffe
Inhoffenstraße 7
Braunschweig 38124
Germany

Re: Spectrum02743-23R1 (Metabolomics insights into the polyketide-lactones produced by *Diaporthe caliensis* sp. nov., an endophyte of the medicinal plant *Otoba gracilipes*)

Dear Dr. Frank Surup:

Your manuscript has been accepted, and I am forwarding it to the ASM Journals Department for publication. You will be notified when your proofs are ready to be viewed.

Sincerely,

Sacha Pidot
Editor, Microbiology Spectrum
